# Time to Lift up COVID-19 Restrictions? Public Support towards Living with the Virus Policy and Associated Factors among Hong Kong General Public

**DOI:** 10.3390/ijerph20042989

**Published:** 2023-02-08

**Authors:** Phoenix K. H. Mo, Yanqiu Yu, Mason M. C. Lau, Rachel H. Y. Ling, Joseph T. F. Lau

**Affiliations:** 1Centre for Health Behaviours Research, Jockey Club School of Public Health and Primary Care, The Chinese University of Hong Kong, Hong Kong SAR, China; 2School of Public Health, Fudan University, Shanghai 200030, China; 3Zhejiang Provincial Clinical Research Center for Mental Disorders, The Affiliated Wenzhou Kangning Hospital, Wenzhou Medical University, Wenzhou 325000, China; 4School of Mental Health, Wenzhou Medical University, Wenzhou 325000, China; 5School of Public Health, Zhejiang University, Hangzhou 311100, China

**Keywords:** COVID-19, living with the virus, resilient coping, self-efficacy, emotional distress, policy support

## Abstract

The fifth wave of the COVID-19 pandemic has caused an unprecedented toll on Hong Kong. As more countries are starting to lift COVID-19 restrictions, it would be important to understand the public attitudes towards lifting COVID-19 restrictions and to identify its associated factors. The present study examined the level of support towards the living with the virus (LWV) policy for COVID-19 among the public in Hong Kong and to identify the associations between resilient coping, self-efficacy and emotional distress with support towards the LWV policy. A random population-based telephone survey was conducted among 500 Hong Kong Chinese adults from 7 March to 19 April 2022, i.e., during the fifth wave of COVID-19 outbreak. Of the respondents, 39.6% showed a supportive attitude towards the LWV policy. Results from the structural equational modeling showed a positive correlation between resilient coping and self-efficacy. Resilient coping was associated with support towards the LWV policy directly and indirectly through a lower level of emotional distress. Self-efficacy had a direct association with support towards the LWV policy but its indirect association through emotional distress was not significant. Interventions that foster resilient coping and self-efficacy would be effective in reducing public emotional distress and promoting their positive view towards the LWV policy.

## 1. Introduction

The COVID-19 pandemic has resulted in an unprecedented impact worldwide. As the economic and social costs caused by the pandemic have been substantial [1], countries have adopted various strategies to mitigate the negative effects of COVID-19. In the beginning of the pandemic, most countries introduced a series of lockdown and social distancing practices to slow down the spread of the pandemic [2]. These polices have varied in stringency across countries and have substantially altered people’s lives in various aspects [3,4,5].

After three years of numerous waves of the COVID-19 pandemic, pandemic fatigue has been reported in various populations [6,7]. At the same time, the increase in COVID-19 vaccination rate and the relatively mild nature of the recent Omicron variant have generated socio-political and economic pressures to open-up and remove social restrictions [8,9,10,11]. Balancing between lowering the infection rate, maintaining individual freedom and facilitating economic production amidst the pandemic has become a significant challenge for policy makers [11]. Two major schools of thoughts in public health policies to deal with the pandemic have thus emerged, the “living with the virus (LWV)” for one and the “zero-virus” strategy for another. There are heated debates on the adoption of either one of them, each with a different rationale [8,12]. The present study investigated a timely and important topic on public attitudes towards the LWV policy in Hong Kong and to examine the roles of resilient coping, self-efficacy and emotional distress in shaping their support towards the policy.

### 1.1. COVID-19 in Hong Kong and the Call for a LWV Policy

Since the beginning of the pandemic, Hong Kong has adopted the zero-virus policy pursued by mainland China. In contrast to the LWV policy, the zero-virus policy involves using numerous public health measures, including mass testing, contact tracing and social restriction measures to stop the community transmission of COVID-19 as soon as it is detected, with the goal of completely eliminating infections and maintaining normal economic and social activities. Following the zero-virus policy, numerous social distancing measures have been introduced to restrict individuals’ movements and activities in Hong Kong. For example, travelers going to Hong Kong need to undergo compulsory quarantine and flights from high-risk countries have been suspended. Individuals who are tested positive or are in close contact with COVID-19 patients need to undergo mandatory confinement in isolation facilities. Schools were suspended and social and public venues, such as gymnasiums, swimming pools, parks and cinemas were closed. The operating time of restaurants has also been significantly restricted. These measures have curbed the spread of the virus significantly in the early phase of the pandemic. There were only a total of 12,649 COVID-19 positive cases in Hong Kong as of 31 December 2021 [13].

However, a drastic change in the pandemic was observed since the highly infectious variant, Omicron, was detected in Hong Kong in January 2022. Cases of COVID-19 started to surge. Between 1 March to 15 March 2022, 409,254 new COVID-19-positive cases were found, which was more than 30 times of the total number of COVID-19 cases since the beginning of the pandemic [13]. The surge represented Hong Kong’s fifth wave of COVID-19 [14]. As of 25 December 2022, a total of 2,463,557 COVID-19-positive cases were reported in Hong Kong [13]. The fifth wave of pandemic has heavily overwhelmed the healthcare system and testing and quarantine facilities of Hong Kong [14]. During the peak of the pandemic, some individuals who tested positive were not provided with timely access to health services and they had to stay at home for self-quarantine. The sweeping number of COVID-19 infections, together with the increased transmissibility of the Omicron variant, have led the government and general public to think that complete elimination measures might not be possible. The usefulness and sustainability of the zero-COVID policy has thus been questioned. The huge economic loss caused by the severe travel restrictions has also called for a need for the authorities to rethink the suitability of the pandemic control measures [15].

Seemingly, increasing public support has been given to the LWV strategy to deal with the pandemic and individuals were increasingly willing to bear the potential costs incurred by the strategy. This is supported by a public survey that has documented a significant decrease in public support on quarantine policy during the fifth wave of the pandemic in Hong Kong. For example, the number of people who supported quarantining anyone who has been in contact with an infected individual dropped from 59% in December 2021 to 45% in April 2022, and those who supported quarantining any location that an infected individual has been in dropped from 45% in December 2021 to 32% in April 2022 [16]. As Hong Kong has started to regain control of the outbreak, it would be an ideal time to reconsider the two different approaches. Understanding the public opinion on the LWV policy would be important to enable effective implementation.

### 1.2. Factors Associated with Attitudes towards COVID-19 Policy

The government’s pandemic mitigation strategies have raised many opinions due to its significant effect on people’s life. Maintaining public confidence and support towards government policy is imperative during a pandemic, as trust and confidence towards the government’s pandemic mitigation measures were related to increased compliance with COVID-19 public health measures [17,18]. Previous studies have reported a marked difference in public opinion on the government’s handling of the COVID-19 pandemic [19,20] and their level of support on government’s action to cope with the pandemic [16,21]. Understanding public perceptions on the government’s response to COVID-19 warrants further investigation.

There is extensive evidence that perceptions and experiences of the disease, personality and psychological factors can help explain individuals’ attitudes towards a health initiative during a pandemic [22,23]. Currently, most of the extant literature on attitudes towards government COVID-19 measures focused on public support of lockdown or control measures aiming towards reducing the spread of the disease and are mostly based on socio-demographic and cognitive factors. For example, a study among Chinese participants reported that factors from the Theory of Planned Behavior (i.e., attitudes, subjective norm, perceived behavioral control) were associated with public attitudes towards COVID-19 lockdown policy, illustrating the important role of risk perceptions in policy support [24]. On the other hand, greater belief in COVID-19 conspiracy theories [25], lower income and those whose income or job opportunities were affected by such measures were more likely to show oppositions to the restriction measures [26,27]. To date, only one study on public attitudes on easing COVID-19 restrictions, conducted in the U.S., could be located [28], showing that COVID-19 experiences and beliefs, political views, consideration of others, as well as long-term orientation, avoidance of uncertainty, reactance and general conspiracy theory beliefs significantly predicted attitudes to easing COVID-19 restrictions [28].

### 1.3. Resilient Coping and Support towards the LWV Policy

The COVID-19 pandemic is an adverse situation that evokes intense negative emotions. Resilience, which refers to the ability to bounce back from adversity, is an important concept for adaptation to the pandemic [29]. Resilience is a dynamic process where individuals adapt positively despite the adversity they experience [30]. According to the Resilience Framework developed by Polk [31], resilience can be characterized into four patterns, namely, dispositional patterns, relational patterns, philosophical patterns and situational patterns. The situational pattern corresponds to the concept of resilient coping, which is manifested as the tendency to use cognitive skills and problem-solving ability effectively to manage stressful situations. It incorporates skills in setting up realistic goals and ability to evaluate consequences of actions and active problem-solving with flexibility, perseverance and resourcefulness. Understanding the resilient coping process is of great interest as it has been found to be important factors of positive psychosocial and physical health outcomes, such as lower cortisol level, higher social support and better perceived health and mental health [32,33,34]. In the context of COVID-19, LWV would imply releasing social restrictions and allowing the public to manage their own level of risk. It would allow more freedom for the general population while carrying a reasonable increase in the level of risk of infection. Individuals who use resilient coping might show more support to the LWV policy as they tend to be more optimistic about the pandemic, show more flexibility in coping with the challenges brought by the increase in risk and are willing to bear the uncertainties and possible costs.

### 1.4. Self-Efficacy and Support towards the LWV Policy

Self-efficacy refers to individual’s belief in his or her capacity to perform behaviors necessary to overcome challenges or produce specific results [35]. It serves as important foundation for motivation, well-being and personal accomplishment [35] and is associated with various positive outcomes such as better mental health and positive health behaviors [36,37]. Studies during the COVID-19 pandemic have demonstrated that stronger self-confidence was significantly associated with lower risks of anxiety and depression and adherence to COVID-19 preventive behaviors [38,39]. No study to date has explored the association between self-efficacy and attitudes towards the LWV policy. Individuals with a higher level of self-efficacy would be more likely to believe that they would be able to cope with the challenges and uncertainties caused by the introduction of the LWV policy, therefore showing higher level of support towards the policy.

Furthermore, self-efficacy can be considered an important dimension of dispositional pattern of resilience, which refers to physical and ego-related psychosocial attributes that serve as protective factors [31]. According to the Polk’s Resilience Framework, dispositional and situational patterns of resilience are mutually reinforcing [31]. Resilient coping may facilitate positive adaptation in the face of adversity and involves reliance on dispositional resources (i.e., self-efficacy). Therefore, it is conjectured that self-efficacy would be correlated with resilience which, would both be associated with lower levels of mental distress and higher levels of support towards the LWV policy.

### 1.5. Emotional Distress and Support towards the LWV Policy

The importance of emotional factors in pandemic control has been widely demonstrated in the literature. Mental distress is related to poorer adherence to health behaviors, avoidance and lower perceptions of control in face of pandemic [38,40,41]. However, the role of emotional distress on attitudes towards pandemic control has received little attention. Negative emotions, such as emotional distress, may prompt individuals to perceive a stressful situation as uncertain or less controllable. Individuals with emotional distress may have a more pessimistic view on the control of the pandemic, perceive themselves as more vulnerable to the infection and are less likely to believe that they are able to deal with the uncertainties brought by the easing of pandemic restrictions. One study in the U.S. found that individuals who felt that their mental health was compromised by COVID-19 showed less support towards the easing of COVID-19 restrictions [28].

The present study examined the level of support towards the LWV policy for COVID-19 among the public in Hong Kong to identify the associations between resilient coping, self-efficacy, emotional distress and support towards the LWV policy. It was hypothesized that resilient coping would be correlated with self-efficacy and both of which would be associated with support towards the LWV policy, directly and indirectly through lower levels of emotional distress. The mediation effects of the association between resilient coping and support towards the LWV policy via self-efficacy/emotional distress were hence tested in the present study. In addition, invariance testing via multi-group SEM was conducted to see if the direction/strength of the structural/indirect paths differed according to COVID-19 ever-infected status.

## 2. Materials and Methods

### 2.1. Participants and Data Collection

A random population-based telephone survey was conducted among Hong Kong Chinese adults aged ≥18 years from 7 March to 19 April 2022, i.e., during the 5th wave of COVID-19 outbreak. By the time of the survey, Hong Kong was still adopting the zero-COVID strategy despite the sharp increase in the number of COVID-19 infections. Household telephone numbers were randomly drawn from the updated landline telephone directories. As this would only be able to reach working individuals after working hours, the 15 min interviews were made by experienced interviewers from 5 pm to 10 pm, in order to avoid over-sampling non-working individuals. The household member whose birthday was closest to the interview date was invited to join the study. Unanswered telephone calls were given at least three attempts before being classified as invalid numbers. Unavailable eligible participants were contacted again by appointments. No incentives were given to the participants. Verbal informed consent was obtained from the participants; the interviewers were asked to sign a form pledging that they had completed the required consent procedures. The ethics approval was obtained from the Survey and Behavioral Research Ethics Committee of the corresponding author’s affiliated institution (No. SBRE-21-0555A).

A total of 500 valid interviews were conducted. The response rate, defined as the number of completed interviews (n = 500) divided by the number of eligible contacts (n = 957), was 52.2%.

### 2.2. Measurements

#### 2.2.1. Background Information

This information included gender, age (years), educational level, marital status, employment status (e.g., full-time and part-time employment) and having any diagnosed chronic diseases (e.g., hypertension, diabetes, pulmonary lung diseases, heart diseases, cerebrovascular diseases, dementia, liver diseases and tumors).

#### 2.2.2. Level of Support towards the LWV Policy

One item assessed participants’ levels of support towards the LWV policy based on the current COVID-19 situation in Hong Kong; the LWV policy included lifting the COVID-19 control policies on social distancing, compulsory facemask wearing, free COVID-19 testing, travel restriction and quarantine. The item was rated on a 5-point Likert scale (1 = extremely low to 5 = extremely high), which was recoded into a binary dependent variable [1 = Supportive (high/extremely high) vs. 0 = Unsupportive (moderate/low/extremely low)].

#### 2.2.3. Resilient Coping

This was measured by the 4-item Brief Resilient Coping Scale (BRCS) that aimed to capture the tendency to cope with stress in an effective, flexible and committed ways [42]. The Chinese version of the BRCS has been validated among university students and showed acceptable construct validity and external validity [43]. Sample items included, “I look for creative ways to alter difficult situations” and “Regardless of what happens to me, I believe I can control my reaction to it”. The items were rated on 5-point Likert scales (1 = does not describe me at all to 5 = describes me very well). The Cronbach’s alpha of the scale was 0.87 in this study.

#### 2.2.4. Self-Efficacy

This was assessed by the item, “I believe I can succeed at most of any endeavours to which I set my mind”, which was selected from the commonly used 7-item General Self-efficacy Scale (GSES) [44]. The item was rated on a 5-point Likert scale (1 = strongly disagree to 5 = strongly agree). A previous scale comparison study found that this single item demonstrated consistently satisfactory reliability and validity compared to the GSES [45].

#### 2.2.5. Emotional Distress Related to COVID-19

A two-item scale was used to assess the extent to which the participants felt (a) panicked and (b) anxious due to the current COVID-19 threat in Hong Kong. The items were rated on a 5-point Likert scale (1 = extremely low to 5 = extremely high). The scale has been commonly used in previous studies investigating emotional responses to emerging infectious diseases (e.g., influenza and COVID-19) [46]. The Cronbach’s alpha of the scale was 0.95 in this study.

#### 2.2.6. Previous COVID-19 Infection

Participants were asked about whether they have ever had COVID-19 positive results of the Nucleic Acid Amplification Test (NAAT) or Rapid Antigen Test (RAT) (yes or no response options).

### 2.3. Statistical Analysis

Supportive attitude towards the LWV policy was used as the binary dependent variable in this study. The differences in background variables between those who were supportive and unsupportive towards the LWV policy were examined by using Chi-square test. Pearson correlation coefficients were derived among resilient coping, self-efficacy and emotional distress due to COVID-19. Univariate and multivariate (adjusted for background factors) logistic regression analyses were conducted to test factors of supportive attitude towards the LWV policy; crude odds ratio (ORc), adjusted odds ratio (ORa) and respective 95% confidence intervals (CIs) were generated. Structural equation modeling (SEM) was fit to investigate the mediation effect of emotional distress due to COVID-19 between resilient coping/self-efficacy and the supportive attitude; the Weighted Least Square Mean and Variance (WLSMV) estimator was used. Two latent variables of resilient coping and emotional distress due to COVID-19 were created from the four and two original scale items, respectively. Multigroup SEM by previous COVID-19 infection was further fit to examine the significance of each structural/indirect paths of the above SEM model between those having and not having a previous COVID-19 infection. A series of models, each constraining a specific individual path (e.g., resilient coping → emotional distress due to COVID-19), were compared to the unconstrained model testing all the paths freely. *p* value > 0.05 in the Wald test would denote between-group invariance on that tested structural/indirect path. The SEM was conducted by using Mplus 7.0, and the other analyses by using SPSS 23.0. Statistical significance was defined as two-sided *p* < 0.05.

## 3. Results

### 3.1. Descriptive Statistics

Of all the participants, 67.0% were female; 69.5% were currently married; 41.2% had a full-time job (41.2%); 25.2% attended college/universities. About one-third aged >60 years (33.4) and had a chronic disease such as hypertension or diabetes (35.8%). The prevalence of having a COVID-19-positive result according to NAAT or RAT was 16.9% (Table 1).

About 40% (39.6%) showed a supportive attitude towards the LWV policy. Those showing a supportive attitude were significantly more likely to be younger (aged 18–30: 20.2% vs. 10.3%), had an educational level of college or above (34.5% vs. 19.0%), in full-time employment (50.5% vs. 35.8%), had a history of COVID-19 infection (23.2% vs. 12.6%) and were less likely to be married (63.6% vs. 73.3%) than those with an unsupportive attitude. No significant between-group differences were found in gender and chronic disease status (Table 1).

### 3.2. Correlation Analysis

Resilient coping was positively correlated with self-efficacy (*r* = 0.49, *p* < 0.001) and negatively correlated with emotional distress due to COVID-19 (*r* = −0.18, *p* < 0.001). The correlation between self-efficacy and emotional distress was statistically non-significant (*r* = −0.04, *p* = 0.378) (See Table 2).

### 3.3. Factors of Supportive Attitude towards the LWV Policy

Consistent with the results of univariate logistic regression analyses, with adjustment of the background factors, multivariate logistic regression analyses showed that resilient coping (ORa = 1.18, 95% CI: 1.08, 1.28) and self-efficacy (ORa = 1.54, 95% CI: 1.16, 2.07) were positively associated with supportive attitude towards the LWV policy, while emotional distress due to COVID-19 was negatively associated with the supportive attitude (ORa = 0.89, 95% CI: 0.82, 0.97). (See Table 3)

### 3.4. Structural Equation Model for Support towards the LWV Policy

Figure 1 presents the SEM testing the mediation effect of emotional distress due to COVID-19 between resilient coping/self-efficacy and support towards the LWV policy. The model showed satisfactory model fit index (χ^2^/*df* = 111.24/82 = 1.36 (*p* of χ^2^ = 0.018); CFI = 0.99; TLI = 0.99; RMSEA = 0.03) with factor loadings of the two latent variables ranging from 0.77 to 0.94 (all *p* < 0.001).

It is seen from Figure 1 that resilient coping showed a significant indirect effect via emotional distress due to COVID-19 on supportive attitude towards the LWV policy (*β* = 0.07; *p* < 0.001), i.e., resilient coping was negatively associated with emotional distress due to COVID-19 (*β* = −0.17; *p* < 0.001), which was in turn negatively associated with the supportive attitude (*β* = −0.40; *p* < 0.001); the mediation effect size was 31.7%. Furthermore, the direct effect of resilient coping on the supportive attitude was significant (*β* = 0.15; *p* = 0.003), indicating a partial mediation between resilient coping and support towards the LWV policy via emotional distress due to COVID-19. In addition, self-efficacy had a significant direct effect on supportive attitude towards the LWV policy (*β* = 0.12; *p* = 0.036), but its indirect effect via emotional distress due to COVID-19 was statistically non-significant (*β* = 0.02; *p* = 0.340), indicating a full mediation effect.

### 3.5. Invariance Testing by Previous COVID-19 Infection Status

All the directions and magnitude of the paths shown Table 4 were invariant regarding positive and negative previous COVID-19 infection statuses.

## 4. Discussion

The pandemic of COVID-19 has disrupted lives across the globe. The implementation of restrictions to reduce the spread of the COVID-19 has sparked considerable debates. As more people are exposed to the less severe Omicron variant, which has become the dominant COVID-19 variant, a shift in public attitudes towards lifting restrictions and living with COVID-19 and calls to ease restrictions and eliminating physical distancing measures have been observed in many countries. However, research on people’s attitudes towards easing COVID-19 restrictions remains scarce.

It is worthwhile to note that a substantial proportion (39%) of the participants showed support towards the LWV policy. The continual mitigation strategies to restrict movements and detection and isolation of those tested positive might have caused tremendous stress and disruption to people’s lives [3,4,5]. Given the scientific evidence about the milder effects of Omicron variant compared to the Delta variant and the fact that the COVID-19 pandemic would not abate in the near future [47], an increasing number of people might believe that their life should return to normal and the restrictions implemented to thwart the spread of the infection must be eased. Indeed, at the time of data collection, Hong Kong has been heavily battered by the fifth, and possibly the worst, local wave of the pandemic. The idea of complete elimination of the virus might have been seen as impossible and LWV may be seen as a more practical and sustainable goal.

Previous studies on resilient coping have been focused on health outcomes [32,33,34]. The present study has extended the current literature and showed that resilient coping was associated with support towards the LWV policy directly and indirectly through lowering emotional distress. Individuals who exhibit resilient coping patterns tend to use adaptive coping strategies to cope with change and stress and plan in a flexible manner despite the stressful circumstance. They might be more optimistic than others regarding the pandemic and more able to extract positive changes from the experiences. They also have a higher ability to achieve positive adaptation despite high level of stress caused by the pandemic, therefore showing higher level of support to the LWV policy.

Self-efficacy is an important dispositional dimension of resilience that would promote positive adaptation. While the relationship between self-efficacy and positive health outcomes has been established in the literature [36,37], findings of the present study extended the extant literature by showing that self-efficacy was correlated with resilient coping and was associated with support of the LWV policy. Findings also lend support to Polk’s Resilience Framework that dispositional (i.e., self-efficacy) and situational (i.e., resilient coping) dimensions of resilience were mutually reinforcing and can both serve to promote positive outcomes [31]. It is plausible that people who are high in self-efficacy believe that they can succeed at most of their endeavors. They are thus more confident that they would manage the possible negative sequels and are more likely to agree with the need to move towards easing the restrictions. However, it is intriguing that the indirect association between self-efficacy and support towards the LWV policy was not significant. More studies are needed to identify other possible mediations for this association.

Despite the extensive discussions on the mental impact of pandemic mitigation measures (such as lockdown policies) [48,49], how mental health affects individual’s attitudes towards such measures has received relatively little investigation. Findings of the present study showed that emotional distress was associated with lower level of support towards the LWV policy, implying that that support towards the LWV policy can be attributable by mental resources. Studies were in line with previous findings that documented an association between poorer mental health and lower level of support in easing COVID-19 restriction [28]. Individuals with emotional distress due to the pandemic might exhibit anxiety that leads to desire to avoid risk of infection. It is therefore conceivable that their mental distress and worry about the pandemic have overridden their desire to go back to normal. Individuals with emotional distress due to the pandemic might also show higher perceived vulnerability to the COVID-19 pandemic, more concern about lack of personal and social resources and the long-term consequences and disruptions on health care systems that the pandemic would cause. They might thus be more likely to believe that suppression measures are needed to drive infection down to levels that make it more easily contained.

It is possible that the COVID-19 pandemic will continue and various kinds of pandemic control strategies may continue to be in place. Some countries might have to reinstall some removed measures if more infectious/severe variants emerge. The present study is one of the few studies in understanding the level of public support towards the LWV policy. Findings on the public’s view towards the easing of COVID-19 restrictions could inform the policymakers to design effective control and health communication strategies in response to further COVID-19 and other outbreaks.

The present study has provided a more nuanced understanding of the role of psychological factors on policy support and its findings have underscored the importance of promoting resilient coping and self-efficacy as means to reduce emotional distress and stimulate public support towards the LWV policy during the pandemic. Indeed, promoting resilience and self-efficacy have been regarded as one of the 10 considerations for effectively managing the COVID-19 transition [50]. It is also evident that resilient coping is influenced by factors from different levels. Interventions and health promotion that promote resilient coping should seek to take a multilevel approach. For example, on the individual level, cognitive or psychological interventions that promote self-regulation, problem-solving, motivation to adapt, persistence and competence are potential strategies in bolstering resilient coping. One study has shown that a brief video that aimed to increase knowledge and understanding of the COVID-19 pandemic and strengthen capacities and competencies in coping with COVID-19 resulted in a significant increase in resilience [51]. On the interpersonal and system level, promoting social support and positive social relationships are also found to be effective in promoting resilient coping [52]. Helping individuals to gain a sense of social connectedness with the society and encouraging social and community participation can increase the resources and perceived competence of the general public in overcoming adversity, thereby promoting resilience.

Furthermore, interventions should also empower the general public by promoting their self-efficacy during the pandemic. Public health interventions should address the general public’s confidence and belief that they have the ability to prevent themselves from the infection and cope with the stressors caused by the pandemic. Social modeling is also another effective way to improve self-efficacy as it allows individuals to observe and learn strategies for overcoming obstacles from others [35,53]. Studies also emphasized the importance of community in influencing public self-efficacy. Community programs can be designed to improve the public’s ability to protect against, respond to and recover from the threats brought by the pandemic, through various levels such as public education, strengthening physical and social infrastructure, promoting public engagement and collaboration between community organizations [11].

The present study is subject to several limitations. The study was cross-sectional in nature so no causality could be assumed. Compared to the Hong Kong general population [54], the age distribution of participants in this study was comparable but more females were recruited. In addition, as age was adjusted for in the models as a categorial variable instead of a continuous variable, residual confounding might exist. The response rate of the present study was only 52.2%, but it was comparable to other local population-based telephone surveys [55,56]. Notably, as we could not interview participants refusing to join this survey, characteristics between participants and non-participants could not be made. Social desirability bias might exist as the Hong Kong government was officially adopting the ‘zero-COVID-19′ policy during the time of the survey. Self-reported measures were used and level of support towards the LWV policy was assessed by a single and self-developed item. Furthermore, due to the variations in COVID-19 pandemic levels and public attitude towards COVID-19 control measures, generalization of the results to other countries should be made cautiously, and international comparison studies are warranted. Finally, the mediating role of emotional distress between self-efficacy and support towards the LWV policy was not significant, so other potential mediators should be considered.

## 5. Conclusions

The present study revealed that during the fifth wave of the COVID-19 pandemic, more than one-third of the general public in Hong Kong were supportive of the LWV policy and their support was motivated by resilient coping, self-efficacy and lower level of emotional distress. Knowing the view of the general public and factors associated with their support will inform government of the feasibility of a similar policy for infection outbreak. Interventions that foster resilient coping and self-efficacy would be effective in reducing public emotional distress and promoting their positive view towards the LWV policy.

## Figures and Tables

**Figure 1 ijerph-20-02989-f001:**
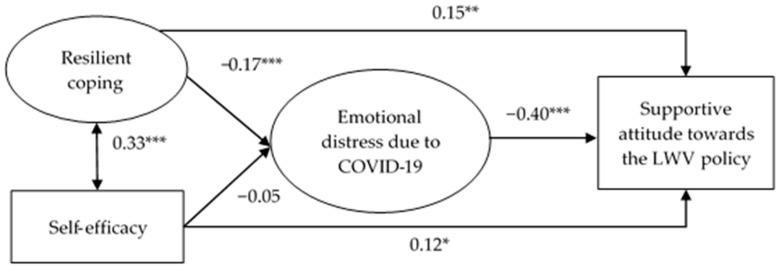
Structural equation modeling (LWV = Living with the virus; * *p* < 0.05; ** *p* < 0.01; *** *p* < 0.001). Standardized coefficients were reported. The model was adjusted for background factors, including gender, age, educational level, marital status, employment status and chronic disease status.

**Table 1 ijerph-20-02989-t001:** Participant characteristics (n = 500).

	Overall	Supportive Attitude towards the LWV Policy		
		Supportive	Unsupportive	*p* Value
	N (%)	n (%)	n (%)	
Overall	500 (100.0)	198 (39.6)	302 (60.4)	
**Background factors**				
Gender				0.092
Female	337 (67.0)	124 (62.6)	211 (69.9)	
Male	165 (33.0)	74 (37.4)	91 (30.1)	
Age groups (years)				0.003
18–30	71 (14.2)	40 (20.2)	31 (10.3)	
31–60	262 (52.4)	103 (52.0)	159 (52.6)	
>60	167 (33.4)	55 (27.8)	112 (37.1)	
Educational level				<0.001
Below college	365 (74.8)	127 (65.5)	238 (81.0)	
College or above	123 (25.2)	67 (34.5)	56 (19.0)	
Marital status				0.021
Others	151 (30.5)	71 (36.4)	80 (26.7)	
Married	344 (69.5)	124 (63.6)	220 (73.3)	
Employment status				0.033
Full-time	208 (41.6)	100 (50.5)	108 (35.8)	
Part-time	41 (8.2)	11 (5.6)	30 (9.9)	
Retired	114 (22.8)	41 (20.7)	73 (24.2)	
Under-employed	27 (5.4)	10 (5.1)	17 (5.6)	
Homemaker	94 (18.8)	30 (15.2)	64 (21.2)	
Others	16 (3.2)	6 (3.0)	10 (3.3)	
Chronic disease status				0.459
No/unknown	321 (64.2)	131 (66.2)	190 (62.9)	
Yes	179 (35.8)	67 (33.8)	112 (37.1)	
**COVID-19 ever infection**				0.002
No/Don’t know/Refused to answer	416 (83.2)	152 (76.8)	264 (87.4)	
Yes	84 (16.8)	46 (23.2)	38 (12.6)	

Note: LWV = Living with the virus. Missing data were excluded from data analysis.

**Table 2 ijerph-20-02989-t002:** Descriptive statistics and correlation matrix of the key variables (n = 500).

	Range	Mean, SD	1	2	3
1. Resilient coping	4–20	15.5, 2.6	-		
2. Self-efficacy	1–5	3.9, 0.7	0.49 ***	-	
3. Emotional distress due to COVID-19	2–10	4.4, 2.4	−0.18 ***	−0.04	-

Note: SD = Standard deviation; *** *p* < 0.001.

**Table 3 ijerph-20-02989-t003:** Factors of support towards the LWV policy (n = 500).

	ORc (95% CI)	ORa (95% CI)
Resilient coping	1.16 (1.07, 1.25) ***	1.18 (1.08, 1.28) ***
Self-efficacy	1.43 (1.09, 1.87) *	1.54 (1.16, 2.07) **
Emotional distress due to COVID-19	0.87 (0.81, 0.94) **	0.89 (0.82, 0.97) **

Note: LWV = Living with the virus; ORc = Crude odds ratio; CI = Confidence interval; ORa = Adjusted odds ratio. *, *p* < 0.05; **, *p* < 0.01; ***, *p* < 0.001. The models were adjusted for background factors, including gender, age, educational level, marital status, employment status and chronic disease status.

**Table 4 ijerph-20-02989-t004:** Invariance testing by previous COVID-19 infection status.

	Previous COVID-19 Infection = No		Previous COVID-19 Infection = Yes		Wald Test			
	*β*	*p*	*β*	*p*	*Estimate*	*p*	Invariance	Model Fit
** *Structural path* **								
Resilient coping → Emotional distress due to COVID-19	−0.13	<0.001	−0.14	0.805	0.17	0.682	**Supported**	χ^2^/*df* = 196.93/172 = 1.15 (*p* = 0.093); CFI = 0.99; TLI = 0.99; RMSEA = 0.02
Self-efficacy → Emotional distress due to COVID-19	−0.11	<0.001	−0.06	0.842	0.42	0.519	**Supported**	χ^2^/*df* = 199.20/174 = 1.15 (*p* = 0.092); CFI = 0.99; TLI = 0.99; RMSEA = 0.02
Resilient coping → Supportive attitude towards the LWV policy	−0.07	0.004	−0.06	0.299	0.31	0.580	**Supported**	χ^2^/*df* = 234.48/174 = 1.35 (*p* = 0.002); CFI = 0.98; TLI = 0.98; RMSEA = 0.04
Self-efficacy → Supportive attitude towards the LWV policy	0.35	<0.001	0.14	0.286	0.89	0.346	**Supported**	χ^2^/*df* = 260.78/176 = 1.48 (*p* < 0.001); CFI = 0.98; TLI = 0.97; RMSEA = 0.05
Emotional distress due to COVID-19 → Supportive attitude towards the LWV policy	−0.42	<0.001	−0.23	0.832	0.14	0.704	**Supported**	χ^2^/*df* = 196.93/172 = 1.15 (*p* = 0.093); CFI = 0.99; TLI = 0.99; RMSEA = 0.02
** *Indirect path* **								
Resilient coping → Emotional distress due to COVID-19 → Supportive attitude towards the LWV policy	0.05	0.001	0.03	0.350	0.01	0.913	**Supported**	χ^2^/*df* = 196.93/172 = 1.15 (*p* = 0.093); CFI = 0.99; TLI = 0.99; RMSEA = 0.01
Self-efficacy → Emotional distress due to COVID-19 → Supportive attitude towards the LWV policy	0.05	<0.001	0.01	0.397	1.04	0.309	**Supported**	χ^2^/*df* = 199.20/174 = 1.15 (*p* = 0.092); CFI = 0.99; TLI = 0.99; RMSEA = 0.02

Note: LWV = Living with the virus. Standardized coefficients were reported. The model was adjusted for background factors, including gender, age, educational level, marital status, employment status and chronic disease status.

## Data Availability

The data presented in this study are available on request from the corresponding author. The data are not publicly available due to respondents’ privacy.

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
