# Peer review of "Time to Lift up COVID-19 Restrictions? Public Support towards Living with the Virus Policy and Associated Factors among Hong Kong General Public"

_ijerph, 2023, doi:10.3390/ijerph20042989_

Round 1

Reviewer 1 Report

General comment

The objective of this study is to examine the degree of support for the COVID-19 Living with Virus (LWV) policy among the general population in Hong Kong and to determine the correlation between support for the LWV policy and resilient coping, self-efficacy, and emotional distress. A sample of 500 individuals was surveyed by telephone during the outbreak.

It was established that more than one-third of the Hong Kong public supported the LWV policy and that their support was motivated by resilient coping, self-efficacy, and low levels of emotional distress.

There has been a paucity of research on attitudes toward relaxing COVID-19 restrictions. Although it is essential to take into account the different cultural contexts of different countries when interpreting these findings, this study may be useful for policymakers and other readers as well.

A few specific comments are noted to enhance the clarity of the manuscript.

Major

i) Table 1 should be summarized by placing the dependent variables, Unsupportive and Supportive, in columns and including basic information on Unsupportive and Supportive groups in Table 1. The reader is also interested in whether there is any bias in the basic information between Unsupportive and Supportive people.

Minor

i) In the logistic regression analysis, age is a covariate, but it was originally thought to be a continuous value. Are the covariates adjusted for continuous values? If age is left as a group as a covariate, then consideration should be given to residual confounding. It may be desirable to model age as a covariate with a continuous variable.

ii) Household telephone numbers were randomly drawn from the updated landline telephone directories; the 15-minute interviews were conducted from 5pm to 10pm by experienced interviewers to avoid oversampling non-working individuals. However, it is unclear what criteria were used to exclude oversampling non-working individuals. It would be beneficial to provide more specific information.

Reviewer 2 Report

Manuscript ID: ijerph-2155452 Review

1.   Sentence construction using English all throughout the paper especially the title should be improved.

2.   What is not clear is this: the authors need to explain how successful or not are the two options adapted in hongkong using data. Second, the authors need to verify if the government is able to implement the expected policies and enforce them according to the options mentioned. Then the authors make an evaluation of the extent of success or failure of either options (zero covid vs. the other option). Was the zero covid policy a failure because it cannot be implemented properly? Or was it a failure because there is no way it can be implemented as expected at all by government for some reasons beyond control?

3.   Then the authors also need to clarify if the 2 options being perceived by the participants  are in accordance with the expected policy and enforcement of the option they agree with. The survey might be asking participants whose impressions about an option is different from the actual option enforced by government.

Reviewer 3 Report

I have reviewed the paper from start to finish.

In their study, the authors examined the level of support towards the living with virus (LWV) policy for COVID-19 19 among the public in Hong Kong, to identify the associations between resilient coping, self-efficacy, and emotional distress with support towards LWV policy. A random population-based tele-21 phone survey was conducted among 500 Hong Kong Chinese adults from March 7 to April 19, 2022,  during the 5th wave of COVID-19 outbreak.

Results from the structural equational modeling showed a positive correlation between resilient coping and self-efficacy. Resilient coping was associated with support towards the LWV policy directly and indirectly through the lower level of emotional distress. Self-efficacy had a direct association with support towards the LWv policy, it is indirect association through emotional distress was not significant.

From a merit point of view, the study has great value because knowing the general public's view and factors associated with their support will inform the government of the feasibility of a similar policy for infection outbreaks.

In my opinion, from a methodological point of view, it is a very robust study with a good analysis using the structural equation model.

I think the work has merit for publication. 

Author Response

Thank you very much for the positive comments